# Carbon Emissions and Its Efficiency of Tourist Hotels in China from the Supply Chain Based on the Input–Output Method and Super-SBM Model

**Jing Zhang and Bing Xia ***

Institute of Geographic Sciences and Natural Resources Research, Chinese Academy of Sciences, Beijing 100101, China; zhangjing244@mails.ucas.ac.cn
*   Correspondence: xiab.16b@igsnrr.ac.cn

**Abstract:** After the COVID-19 epidemic, the recovery of tourism growth faced more pressure on carbon emissions. As an important sector of tourism economic recovery, the tourist hotels' carbon emissions cannot be ignored. This study combined the EEIO (the environmentally extended input–output) model and Super-SBM (slacks-based measure) model to measure carbon emissions and its efficiency including indirect carbon emissions from the supply chain in China in 2002–2022. The results indicate that: Tourist hotels in most eastern provinces exhibit the U-shaped pattern in terms of carbon emissions. the majority of indirect carbon emissions from these hotels originate from the food and tobacco processing sectors. Indirect carbon emissions of tourist hotels in recent years might be decoupling with economic growth because of its decreasing trend. The implementation of energy-saving technology can enhance the carbon efficiency of tourist hotels, while regional economy growth can benefit the carbon emissions efficiency indirectly. Environmental pollution control investment and employment population are external driving factors affecting carbon emissions from tourist hotels. The objective of this study is to establish a scientific framework that promotes low-carbon development within the accommodation industry, both in China and comparable regions globally.

**Keywords:** tourist hotels; carbon emissions; supply chain; efficiency; China

## 1. Introduction

According to Global Info Research, global hotel revenue is expected to reach approximately 2876 million USD in 2023 and 380,430 million USD by 2030 [1]. The global hotel and resort market size exceeded 1 trillion USD annually from 2016 to 2019 [2]. However, the negative impact of its carbon emissions cannot be ignored. In 2023, global energy-related carbon emissions increased by 1.1%, by 410 million tons, reaching a historic high of 37.4 billion tons [3], and the accommodation sector is the second-largest source of tourism's carbon emissions after transportation [4]. Therefore, 1% of global total carbon emissions come from hotels, hotels contributed approximately 363 million tons of the 36.3 billion tons of carbon dioxide emitted globally in 2022 [5], and this proportion is in a constantly growing stage [6]. The supply chain of hotels includes a great many intermediate links, which are completely caused by its close correlation with other sectors [7]. The tourist hotels referred to in this study are star rated hotels. Taking the 2014 World Input–Output Table as an example, the product department within the accommodation and food service sector utilizes 12.6% of the outputs from public administration and mandatory social security services, 9.6% of the offerings from headquarters and management consulting activities 7.0% of the construction sectors, and 3.9% of the wholesale trade in the production process. Researchers have demonstrated that indirect carbon emissions, including those from the supply chain, exceed direct emissions by a factor of over four [8]. Hence, the tight integration with other industries has resulted in considerable indirect carbon emissions for hotels, a factor that warrants serious consideration.

In 2023, China accounted for 33.69% of global carbon emissions, reaching 12.60 billion tons, which is higher than the sum of the United States and the European Union [9]. Moreover, as the largest tourism consumption country, China's accommodation industry holds unique characteristics and serves as a representative example [10]. Since 1978, China's accommodation industry has experienced a transformation process from a public-owned to an individual and private economy [11]. China's tourist hotels are transforming into high-quality development [12]. The revenue from China's tourist hotels reached 160.90 billion USD in 2023, contributing 3.28% to the total revenue of the tourism industry [13]. China's tourism showed a positive recovery after the COVID-19 epidemic, and the leisure and vacation share increased sharply [14]. In 2023, the accommodation and catering industry achieved an added value of 2.1 trillion CNY, marking a 14.5% year-on-year growth. This figure surpasses that of the same period in 2019 and stands as the highest growth rate among all industries [15]. During the Spring Festival in 2024, the order of the OTA accommodation industry was significantly more than that of the same period in 2019. However, there is a potential risk that the industry of accommodation could emerge as a significant contributor to China's tourism-related carbon emissions.

The carbon emissions of the accommodation sector have become a major field of concern in more and more regions, such as Spain [16], Thailand [17], Japan [18], Mexico [19], Brazil [20], Indonesia [21], Italy [22], etc. Research in the area of hotel carbon emissions primarily focuses on several aspects. Firstly, it involves calculating the carbon emissions resulting from energy consumption within the accommodation industry [23–25], and certain scholars have identified tourist hotels as the subsector within the accommodation industry that generates the highest carbon emissions [26], due to uncontrolled electricity consumption [23]. Secondly, hotels, especially star-rated ones, significantly contribute to carbon emissions in the tourism sector [27], and the energy consumption and carbon emissions of hotels play a pivotal role in the overall environmental footprint of the tourism industry [28]. Additionally, eco-efficiency analysis reveals tourist hotels are the primary sources of carbon emissions within the tourism sector, particularly due to their supply chain operations [29]. Thirdly, improving energy efficiency, using alternative renewable energy, standardizing electricity use, and indoor greening is regarded as a significant strategy for reducing the carbon emissions of tourist hotels [6,25,30]. In conclusion, the connection between tourist hotels and carbon emissions is substantial, highlighting the need for sustainable practices and emission reduction strategies within the hotel industry to mitigate environmental impacts.

While existing research has made significant advancements, it predominantly focuses on analyzing direct carbon emissions or their driving factors within tourist hotels. However, there is a notable lack of scholars who have measured the indirect carbon emissions of tourist hotels as a distinct ecological economic system, particularly from a supply chain perspective, and pinpointed the sources of these emissions, especially in the context of global climate change during the post-pandemic era. To fill the above knowledge gap, the innovation of this paper is (1) From the viewpoint of the supply chain to combined Super-SBM and EEIO model to construct a research framework of "source-efficiency-driving-decoupling" of tourist hotel' carbon emissions. (2) Tourist hotels are regarded as an ecological-economic system, which includes carbon emissions part and economic growth part. Then the carbon emissions efficiency was estimated. (3) To provide more targeted policy recommendations, this study analyzed the immediate sources of carbon emissions and the factors driving carbon emission efficiency in hotels. Consequently, the research aims of this study are as follows: firstly, to compute the direct and indirect carbon emission efficiency and trends within China's tourist hotel supply chain spanning from 2002 to 2022. Secondly, visualize the direct and indirect carbon emissions to interpret their spatial distribution. Thirdly, to pinpoint the primary sources of indirect carbon emissions emanating from tourist hotels and to investigate the interplay between carbon emissions and the economic value of Chinese tourist hotels. Fourthly, to dissect the carbon efficiency of hotels when considering both direct and indirect carbon emission scenarios. Fifthly, to

delve into the drivers that influence the carbon emissions and efficiency of Chinese tourist hotels. This study takes China as a typical case of the global accommodation industry, aiming to construct a framework for carbon emissions and their impact on the tourism or accommodation industry in similar regions of the world.

The practical significance of this study lies in, firstly, visualizing the geographical differences in carbon emission, pinpointing the primary sources of indirect carbon emissions from Chinese tourist hotels, and suggesting more tailored emission reduction strategies; secondly, identifying the factors that influence the carbon emissions of tourist hotels and proposing a green decoupling development strategy with a stronger emphasis on carbon reduction. The theoretical significance of this study includes, firstly, establishing a measurement system and theoretical framework for assessing direct and indirect carbon emissions of tourist hotels using the input–output method; secondly, offering novel perspectives for evaluating the coordination between the economic development of tourist hotels and the ecological environment; thirdly, this study examines the carbon emissions of Chinese hotels across an extended time series, offering a comparative benchmark for studies that focus on smaller spatial scales within shorter time frames.

## 2. The Literature Review

### 2.1. Carbon Emissions and Efficiency of Hotels

Most activities related to hotels (such as hotel construction, operation, sales, management, transportation, etc.) require fossil fuels or electricity to provide energy. This consumption leads to greenhouse gas emissions, mainly carbon dioxide [31]. In the context of global climate change, an increasing number of countries and researchers have started directing their focus towards studying carbon emissions and efficiency within the hotel industry. According to the Classification and Evaluation of Star Levels for Tourist Hotels standard (GB/T 14308-2023) [32], tourist hotels are capable of providing accommodation facilities with dining and related services to tourist guests on a night basis. The tourist hotels referred to in this study are star rated hotels, which are designated as one-star, two-star, three-star, four-star, and five-star hotels.

Calculating carbon emissions and developing low-carbon strategies were most attractive. Zi Tang [27] has proven that star hotels' carbon emissions increased by 3.84 times from 1998 to 2009 in China. Salehi, M [24] found that luxury hotels are three to four times in Iran higher than that of similar hotels studied in the past, and the carbon intensity is seven times higher. These studies show that the hotel is facing serious carbon emission reduction challenges. However, some research results are optimistic. Jiachen Li [33] insisted that the hotels' carbon emissions have decreased obviously in recent years. In another hand, the researchers found that the indirect carbon emissions ignored in the accounting process often exceeded people's expectations. Jun Liu [34] found that the majority of carbon emissions from the service industry, particularly the accommodation sector, in the studied regions are indirect, constituting 74.99% of the overall emissions. When Cadarso [35] included the civil infrastructure related to hotels, as well as energy such as electronic machinery and transport vehicles into the carbon footprint calculation, it revealed a 34% increase in total carbon emissions. This also shows that measuring the direct carbon emission from the hotel is incomplete. Certain academics have also emphasized the significance of decreasing carbon emissions in the hotel industry for enhancing brand competitiveness and fostering environmental efficiency [36]. Research indicates that, in terms of carbon emissions reduction, international hotel chains exhibit superior energy efficiency and branding value compared to independently operated hotels [37]. It is imperative to strike a balance between economic and social contributions and ecological impacts within the tourist hotel industry, particularly in China, where income and carbon emissions are still intricately linked and have not achieved decoupling [38]. The energy consumption of hotels significantly contributes to their carbon footprint, and strategies to reduce emissions should consider all aspects of hotel operations, not just energy consumption [39].

For the research methods of hotel carbon emissions, experimental observation methods, life cycle methods, etc. have been mostly used in the past. For example, Salem, R [40] found through experimental observation methods that using a new emission reduction method, namely cogeneration, the average reduction percentage of carbon emissions is 36%. Spiller, M [6] used the life cycle model to evaluate the carbon emission cycle of a medium-sized hotel in Greece. For the study of hotel efficiency, data envelopment analysis is often used. Sánchez [41] calculated the efficiency of 52 hotels in the various Spanish provinces and compared them. With the deepening of research, scholars have found that hotels not only achieve an expected output of operating revenue but also unexpected output. To combine with reality, the Super-SBM models that include unexpected output are being increasingly applied by researchers. Deng [42] employed the Super-SBM model to assess hotel efficiency across 31 provinces. This method addresses the limitation of traditional DEA models which fail to differentiate between multiple simultaneously efficient decision units. Consequently, this study adopted the Super-SBM model to evaluate the ecological efficiency of tourist hotels.

*2.2. Input–Output Model for Carbon Emissions*

As early as 2009, Susanne Becken [31] proposed 'top-down' and 'bottom-up' accounting methods for carbon emissions from tourism. Among them, the former calculates carbon emissions based on sub-sectors and elements of consumer terminals, mainly using the life cycle method. However, this method focuses more on the assessment of microenvironmental impacts, making it difficult to obtain huge data support to measure the greenhouse gas emissions caused by a certain industry in a country, and errors in the calculation of indirect carbon emissions result in an underestimation of the overall carbon emissions. In contrast, the latter can more accurately measure carbon emissions from the supply chain. The input–output model is the most commonly used and ideal method for studying industrial structure, which was later extended and applied to the analysis of carbon dioxide and greenhouse gas emissions. Due to its unique industry attributes, hotels require close investment from other industries in order to implement more refined policies. At the current stage, the application of this model is relatively limited, but indirect carbon emissions and sources can be obtained through input–output analysis. Therefore, this model has great potential for future application in the hotel industry. Moreover, the EEIO (environmentally extended input–output) offers a straightforward and reliable approach for assessing the relationship between economic consumption activities and their environmental impacts, as well as the environmental impacts reflected by the flow of services and products between departments. There is a unique advantage in calculating the impact of implicit environmental factors in the upstream sector on downstream sectors and final demand. Therefore, this study utilizes the EEIO method to calculate the carbon emissions of hotels with greater accuracy, including indirect carbon emissions stemming from the supply chain [43].

The input–output model is employed to investigate the interdependencies between various economic sectors within a country. Initially proposed by Leontief in the 1930s, it serves as a valuable instrument for precisely assessing indirect carbon emissions, including those originating from the supply chain [44]. The input–output method has been popularly used in the study of tourism destination carbon footprint in recent years (Whittlesea, E.R), global carbon measurement (Lenzen), and climate policy [45]. In the context of hotels, the EEIO focuses on factors such as the number of guest rooms and the area of the catering department as quasi-fixed inputs. To estimate dynamic productivity, a non-radial Malmquist productivity index is utilized, revealing that chain-operated hotels have surpassed independently operated ones over time in terms of performance [46]. Another research incorporates a multi-component data envelopment analysis and a global assurance region model to assess hotel performance. The findings indicate that hotels managed by international chains outperform those managed by local chains and independently managed hotels [47]. Additionally, a study on Taiwanese tourist hotels suggests that foreign-owned hotels exhibit better meta-efficiency and technology gap compared to domestic hotels,

with more productive employees [48]. The EEIO is commonly applied in assessments, incorporating embodied carbon emissions from the supply chain as a key component. Kurzweil [49] discussed the estimation of direct carbon emissions from Australia's tourism and related activities in 2010. Sharp [50] emphasized the role of indirect carbon emissions when assessing the carbon footprint of Iceland tourists during 2010–2015. In 2017, Liu [34] found that the indirect carbon emissions in the investigated areas contributed up to 75% of all carbon emissions. Xia [29] found that tourist hotels became the largest sector due to generous carbon emissions in the supply chain. The above studies confirm that indirect carbon emissions cannot be ignored.

The existing research has made fruitful achievements on hotel carbon emissions, and the innovation methods have been more diverse. However, there are still several knowledge gaps: Further research is required to investigate the correlation between carbon emissions and the revenue of tourist hotels. Additionally, there is inadequate measurement of carbon emissions within the supply chain of tourist hotels, and the exploration of impact indicators related to the carbon emission.

This research filled the above knowledge gap from a macro perspective in the following aspects: First of all, what are the spatial-temporal features of direct and indirect carbon emissions in the transformation process of China's tourist hotels? Second, considering the input–output relationship between relevant sectors, which industry is the main indirect source of carbon emissions from tourist hotels? Thirdly, what is the coupling relationship between the rapid economic growth of tourist hotels and carbon emissions? Fourthly, under different scenarios, what are the influence factors and mechanisms of tourist hotels' efficiency?

## 3. Methodology

In this study, the EEIO model and carbon emission factors provided by the IPCC (Intergovernmental Panel on Climate Change) were utilized to compute carbon emissions, as outlined in Section 3.1. Coupling Analysis was employed to investigate the link between carbon emissions and revenue in the Chinese tourism industry, while the Super-SBM Non-Oriented Model was used to calculate carbon emission efficiency under various scenarios, as described in Sections 3.2 and 3.3. Furthermore, the Panel Tobit model was applied to explore the internal factors driving carbon emissions in Chinese tourism hotels, and Geographic Detector was used to identify the external driving factors influencing these hotels, as detailed in Section 3.4.

### 3.1. Calculation of Carbon Emissions
3.1.1. Direct Carbon Emissions

The Intergovernmental Panel on Climate Change (IPCC) is utilized to calculate carbon emissions due to its reference approach for obtaining direct household carbon emissions (HCEs) from different types of fossil fuels [51]. The industry's energy emission coefficient is associated with the added value of each sector and the carbon emission coefficient. The detailed steps are:

$$\eta^{th} = \frac{\Sigma_{k=1}^{r}\delta_k \times EC_k^{ac}}{l^{ac}} \tag{1}$$

Among them, $\eta^{th}$ is the energy consumption coefficient of China's tourist hotels, $\delta_k$ is the total energy consumption ($k = 1,2,3... r$), $EC_k^{ac}$ represents the $k$ energy consumption, and $l^{ac}$ is the added value of the accommodation and catering industry. The formula of the direct carbon emission calculation is as follows:

$$CET_{direct}^{th} = u \times \eta^{th} \times TR^{th} \tag{2}$$

Among them, $CET_{direct}^{th}$ represents the direct carbon emissions generated by hotels, $\eta^{th}$ is the energy emission coefficient in Formula (1), and $TR^{th}$ represents the added value

of the tourist hotel. Direct carbon emission can reflect the emission of emission sources directly controlled or owned by enterprises.

### 3.1.2. Total Carbon Emissions and Indirect Carbon Emissions

The overall carbon emissions of tourist hotels are computed using input–output tables from 2002, 2007, 2012, and 2017. These tables illustrate the sources of input and the consumption of output for production activities across various departments within a specified timeframe [52]. More importantly, the input–output table can comprehensively and systematically represent the complete physical movement process of goods or services from production to use in various sectors of the national economy. Therefore, the use of EEIO based on input–output table is beneficial for calculating total and indirect carbon emissions due to their ability to identify coefficients that, when altered slightly, result in significant reductions in industrial greenhouse gas emissions [53].

Table 1 displays the composition of the input–output table, with the red box section highlighting the supply chain within the table. The background is a green quadrant, where the row direction reflects the value of goods or services produced by a certain industry sector for various end uses, and the column direction reflects the scale and composition of each end use. The quadrant with a blue background is used in the middle row direction and in the middle column direction, denoted by $ij$. The orange quadrant in the background reflects the added value and composition of various industrial sectors. The last line represents the total input, which is equal to the total output ($x'^1 + \ldots + x'^i + \ldots + x'^n = x^1 + \ldots + x^i + \ldots + x^n$).

**Table 1.** The structure of the input–output table.

| | | Intermediate Use | | | | | | |
|---|---|---|---|---|---|---|---|---|
| | Industrial Sector | D1 | … | Dj | … | Dn | Final Use | Total Output |
| Intermediate Input | DS1 | $z^{11}$ | … | $z^{1j}$ | … | $z^{1n}$ | $f^1$ | $x^1$ |
| | ⋮ | ⋮ | | ⋮ | | ⋮ | ⋮ | ⋮ |
| | DSi | $z^{i1}$ | … | $z^{ij}$ | … | $z^{in}$ | $f^i$ | $x^i$ |
| | ⋮ | ⋮ | | ⋮ | | ⋮ | ⋮ | ⋮ |
| | DSn | $z^{n1}$ | … | $z^{nj}$ | … | $z^{nn}$ | $f^n$ | $x^n$ |
| Value added | | $q^1$ | … | $q^j$ | … | $q^n$ | | |
| Total input | | $x'^1$ | … | $x'^i$ | … | $x'^n$ | | |

In the input–output table, tourist hotels belong to the accommodation and catering sectors.

$$z^{i \sim th} = \frac{TR^{th}}{f^{ac}} z^{i \sim ac} \tag{3}$$

Among them, $z^{i \sim th}$ represents the middle use of department $i$ in tourist hotels to department $j$, $f^{ac}$ represents the final use, $TR^{th}$ is the added value of tourist hotels, and the intermediate utilization of the accommodation and catering industry towards other sectors is represented by $z^{i \sim ac}$.

$$X = (I - A)^{-1} Y \tag{4}$$

In Formula (4), $X$, $Y$, and $A$, respectively, represent the output final use.

$$X_t = \left[ (I - A)^{-1} \right]^T L \tag{5}$$

where $L$ represents the matrix of the value-added, and the matrix $\left[(I-A)^{-1}\right]^{T}$ is the transposition matrix of $(I-A)^{-1}$. $X_t$ represents the total input in the production of a certain product, then the total carbon emission formula is expressed:

$$CET_{total}^{th} = \mu \times \eta^{th} \times X_t^{th} \tag{6}$$

Indirect carbon emissions in this research are generated from the immediate input sectors in the supply chain [54]. Indirect carbon emission of each sector ($CE^{i\sim th}$) can calculate according to the input proportion, and the calculation formula is as follows:

$$CE^{i\sim th} = \left(CET_{total}^{th} - CET_{direct}^{th}\right) \times \frac{z^{i\sim th}}{l^j} \tag{7}$$

Among them, $CE^{i\sim th}$ represents indirect carbon emissions, $CET_{total}^{th}$ represents complete carbon emissions, $CET_{direct}^{th}$ represents direct carbon emissions, $z^{i\sim th}$ represents the middle use of department $i$ in tourist hotels to department $j$, and $l^j$ represents the added value of department $j$. In this study, the EEIO overcame traditional input–output models and accurately calculated indirect carbon emissions from the hotel supply chain. It lays the foundation for analyzing the relationship between the hotel department and other departments.

### 3.2. Coupling Analysis of Carbon Emissions and Income

Coupling analysis comes from physics [55]. Scholars extended this method for qualitative and quantitative analysis of data, especially favored by economists, for the analysis of multi-element, multi-industry, and multi-system interaction [56]. Coupling is a suitable method for examining the connection between carbon emissions and the economy, as it aids in comprehending the interaction between the digital economy and carbon emissions [57]. This research coupled the income (I) in 2002 and 2022 with direct carbon and indirect carbon emissions (DC, IC), respectively, and divided it into four types, namely, high-carbon and high-income type I (I < 35, DC, IC > 6), high-carbon and low-income type II (I > 35, DC, IC > 6), low-carbon and low-income type III (I < 35, DC, IC < 6), and low-carbon and high-income type IV (I > 35, DC, IC < 6).

### 3.3. Assessment of the Carbon Emission Efficiency: Super-SBM Non-Oriented Model

Tone [58] proposed the SBM in 2002. On this basis, the efficiency obtained from the Super-SBM model allows the efficiency value to be bigger or equal to 1, which solves the problem of complete efficiency of multiple decision-making units and has been applied in many fields [59,60]. The Super-SBM model can identify Pareto-efficient projections and provide accurate super-efficiency scores without overestimation, making it a suitable choice for computational efficiency [59]. By combining the benefits of radial and SBM models, the joint SBM model guarantees Pareto-efficiency and continuous scores, enhancing computational efficiency [61]. Hence, this study assesses the effectiveness of both direct and overall carbon emissions generated by tourist hotels, inclusive of their supply chains, through the establishment of a Super-SBM model using the DEA_Solver5.0. The formula is as follows:

$$s.t. \begin{cases} \min\rho = \left(1 - \frac{1}{m}\sum\limits_{i=1}^{m}\frac{s_i^-}{x_{ik}}\right) \Big/ \left[1 + \frac{1}{q_1+q_2}\left(\sum\limits_{r=1}^{q_1}\frac{s_r^+}{y_{rk}} + \sum\limits_{r=1}^{q_2}\frac{s_t^{b-}}{y_{tk}}\right)\right] \\ x_k = M\lambda + s^-, y_k = N\lambda - s^+, b_k = P\lambda + s^{b-} \\ \lambda \geq 0, s_i^- \geq 0, s_r^+ \geq 0, s_t^{b-} \geq 0 \end{cases} \tag{8}$$

where $\rho$ is the efficiency, that is, the carbon emission efficiency in the direct or total situation. $m$, $q_1$, and $q_2$ represent the quantities of indicators for inputs, desired outputs, and undesired outputs, respectively. $x$, $y$, and $b$ are input, desired and undesired output variables. $M$, $N$, and $P$ are input–output matrixes. It indicates that the DMU is relatively invalid and needs to improve the variables when $0 < \rho < 1$.

*3.4. Driving Factors*

3.4.1. Internal Driving Factor Analysis-Panel Tobit Analysis

James Tobin first put forward the Tobit model in 1958 [62]. The Tobit model was employed to discern the underlying driving forces and pivotal factors influencing various domains, including the economy, society, policy, and climate [63]. This paper applied the StataMP-64 to find the driving factors that influence carbon emissions and its efficiency of the tourist hotel. The model expression is:

$$y_{it}^* = ax_{it} + \varepsilon_{it} \quad y_{it} = \begin{cases} y_{it}^* & , y_{it}^* \geq 0 \\ 0, & y_{it}^* < 0 \end{cases} \quad i = 1, \cdots, N \ and \ t = 1, \cdots, T \ \varepsilon_{it} \sim N\left(0, \sigma^2\right) \quad (9)$$

where $t$ represents the research year, $i$ represents 30 provinces in China, $x_{it}$ represents an independent variable, $y_{it}$ represents variable, and $\varepsilon_{it}$ represents disturbance term.

3.4.2. External Driving Factor Analysis-Geographic Detector

Geographic detector is a method to detect spatial differentiation to reveal the driving force of dependent variables [64]. This study uses the Geographic Detector to identify external driving factors that affect carbon emissions from tourist hotels.

The formula of the factor detector is expressed as follows:

$$q = 1 - \frac{\sum\limits_{h=1}^{L} N_h \partial_h^2}{N\sigma^2} = 1 - \frac{SSW}{SST} \quad (10)$$

$$SSW = \sum_{h=1}^{L} N_h \partial_h^2 \quad (11)$$

$$SST = N\sigma^2 \quad (12)$$

where $h$ is the number of categories that affect the carbon emission factors of tourist hotels, $h \in [1, L]$, $N_h$ is the number of units in layer $h$, and $N$ is the total number of units in the study area, and $\partial_h^2$ and $\sigma^2$ are the variance of a certain layer $h$ and the variance of the entire region, respectively. SSW represents the sum of variances within layers, while SST denotes the total variance across the entire region. $Q \in [0, 1]$, the larger the $q$ value, the more pronounced the hierarchical heterogeneity. Due to the fact that carbon emissions from tourist hotels may be the result of the combined effects of various factors, a higher $q$ value indicates a stronger explanatory power of the influencing factors on carbon emissions from tourist hotels, while the opposite indicates a weaker explanatory power When $q = 1$, it indicates that the influencing factor completely controls the carbon emissions of tourist hotels. When $q = 0$, it suggests that the carbon emissions of tourist hotels remain unaffected by this particular factor.

In this study, $Y$ represents the carbon emissions of tourist hotels, and $X$ represents the influencing factors (a total of 12).

The interaction detector calculates the q(X1) and q(X2) of factors X1 and X2, as well as the q(X1∩X2) of the interaction between X1 and X2. According to the relationship between q(X1), q(X2), and q(X1∩X2), the interaction is measured.

*3.5. Indicators and Data*

3.5.1. Indicators for Carbon Emissions Efficiency Assessments

According to the Cobb–Douglas product function, which measures the operational laws of the economy through labor and capital [65]. In this study, the number of hotel staff and fixed asset investments were utilized as input indicators, while hotel industry revenue served as the output indicator. Additionally, direct and total carbon emissions were considered as undesirable outputs to assess the carbon emission efficiency across various emission scenarios presented in Table 2.

**Table 2.** Indicators for evaluating carbon emissions efficiency.

| Types | Indicators | Unit |
|---|---|---|
| Input | Labor | 10 thousand people |
| | Investment | Million |
| Output | Income | Million |
| Undesirable output | Direct/Total carbon emissions | Ton |

3.5.2. Regression Model Indicator Selection

The established regression model serves to pinpoint the influential factors that determine the carbon emissions of tourist hotels, which will help put forward more targeted carbon emission reduction measures and optimization paths [66]. Existing research has shown that the internal impact of the system can include scale effect, technology effect [67], and structure effect [68]. The scale effect was represented by the operating income (Income), the technical effect was represented by the energy per unit (Energy), and the structural effect was represented by the proportion of five-star hotels in the number of Chinese star hotels (TI). The external factors also can influence the carbon emissions and its efficiency of the hotel. The external factors can include civilization, traffic conditions, urbanization, energy efficiency, economics, resource, etc. [69]. The scale of college students (ST) represented the degree of civilization, the number of highway kilometers (HKM) represented the traffic conditions, the ratio of the urban population (UP) represented urbanization, the per capita gross regional product (GDP) represented the regional economic, and energy consumption per unit of GDP (GPE) served as an indicator of energy efficiency, while per capita water resource availability (GPW) represented resource abundance. Table 3 shows the statistical description results of the variables.

**Table 3.** Descriptive statistical results of driver indicators.

| Type | Variable | Mean | Std. Dev. | Min | Max |
|---|---|---|---|---|---|
| Direct Carbon Emissions | DC | 8.96 | 8.75 | 0.32 | 63.47 |
| Indirect Carbon Emissions | IDC | 7.31 | 8.98 | 0.09 | 83.95 |
| Efficiency of Direct Carbon Emissions | $CET_{direct}$ | 0.78 | 0.32 | 0.17 | 4.01 |
| Efficiency of Total Carbon Emissions | $CET_{total}$ | 0.75 | 0.34 | 0.14 | 4.05 |
| Scale Effect | lnIncome | 12.60 | 0.98 | 9.14 | 14.81 |
| Technical Effect | lnEnergy | 10.51 | 0.75 | 7.38 | 12.79 |
| Structure Effect | TI | 0.37 | 0.17 | 0.00 | 0.84 |
| Regional Economic | lnGDP | 10.39 | 0.81 | 8.09 | 12.15 |
| Urbanization | UP | 0.54 | 0.15 | 0.14 | 0.90 |
| Water Resource | GPW | 7.04 | 1.24 | 3.95 | 9.75 |
| Energy Efficiency | lnGPE | 1.00 | 0.76 | 0.03 | 5.83 |
| Civilization | lnST | 13.26 | 0.90 | 10.01 | 14.85 |
| Traffic Conditions | lnHKM | 9.66 | 4.39 | −3.00 | 12.73 |

Taking the regression model of direct carbon emissions as an example, the model formula is:

$$DC = a_0 + a_1 lnIncome + a_2 lnEnergy + a_3 TI + a_4 lnGDP + a_5 UP + a_6 GPW + a_7 lnGPE + a_8 lnST \\ + a_9 lnHKM + \varepsilon \tag{13}$$

3.5.3. Selection of Geographic Detector Indicators

Due to China's vast territory and abundant resources, there are significant differences in geographical conditions, regulatory frameworks, technological advancements, and consumer behavior changes among provinces. To further accurately investigate the influencing factors that affect carbon emissions from Chinese tourist hotels, this study uses the ArcGIS

10.8 and selects the following 12 indicators (Table 4): five years (2002, 2005, 2010, 2015, 2019, and 2022) and two scenarios (Y1: direct carbon emissions; Y2: indirect carbon emissions) were used to explore the external driving factors of geographic detectors.

**Table 4.** External driving factors of carbon emissions in different contexts.

|  | Indicator | Unit |
|---|---|---|
| X1 | Average altitude | m |
| X2 | Average temperature | °C |
| X3 | Average precipitation | mm |
| X4 | Forest coverage rate | % |
| X5 | Area of protected zone | % |
| X6 | Investment in environmental pollution control | Billion |
| X7 | Employed population | 10 k People |
| X8 | Internal expenditure of R&D funds | 10 k CNY |
| X9 | National technology market transaction volume | 10 k CNY |
| X10 | Jurisdictional area | 10 k Hectares |
| X11 | Number of domestic tourism organizations organized by travel agencies | 10 k People |
| X12 | Overall daily average of energy-saving and emission reduction policy search index | / |

### 3.5.4. Data Sources

The input–output table in 2002, 2007, 2012, and 2017 used in this research are from China's carbon emissions database (www.ceads.net.cn). The per capita GDP were sourced from the official statistical yearbook. The per capita water resources and road mileage are from the Environmental Statistics Yearbook of China, the added value of the accommodation and catering industry comes from the China's National Bureau of Statistics, and the hotel income data comes from the Tourism Statistics Yearbook of China. The income is adjusted for inflation according to China's consumer price index, with 2002 as the base year. The data of temperature, precipitation, and altitude are from the Resource and Environmental Science and Data Center of the Chinese Academy of Sciences (https://www.resdc.cn/DOI/doiList.aspx, accessed on 1 September 2024). The number of domestic tourism organizations organized by travel agencies comes from the Ministry of Culture and Tourism of China, and the search index for energy conservation and emission reduction policies comes from Baidu Index, the largest search engine in China. The number of hotel guests in this study was calculated by multiplying the number of rooms in each province by the room occupancy rate, 2 people, and 365 days.

## 4. Results and Analysis

### 4.1. Carbon Emission Trend

The carbon emissions of Chinese tourist hotels are decreasing, while the revenue of tourist hotels is increasing. In terms of quantity, the success of Beijing's bid for the Olympics in 2003 and Shanghai's bid for the World Expo in 2004 greatly stimulated the explosive growth of China's tourist hotels. The Star Classification and Evaluation of Tourist Hotels (GB/T 14308-2023) released in 2010 strictly enforced the star rating standards for tourist hotels, removing some hotels that did not meet the star rating standards from the list of star-rated hotels, resulting in a reverse V-shaped growth of star rated hotels nationwide. From the perspective of carbon emissions and economic situation, the downward trend of indirect carbon emissions was significantly different from the linear upward trend of five-star hotels (Figure 1). It shows that, the better the services, the lower the indirect carbon emissions of tourist hotels. The average direct carbon emissions increased from 57,500 tons in 2002 to 118,400 tons in 2012 and then decreased to 35,700 tons in 2022. The average indirect carbon emissions of Chinese tourist hotels increased from 69,800 tons in 2002 to 139,900 tons in 2006 and then decreased to 20,800 tons in 2022. Moreover, the indirect carbon emissions from tourist hotels constitute a smaller proportion compared to their direct emissions, a finding that often diverges from the typical carbon emission patterns observed in the

tourism industry [70]. The indirect carbon emissions have had a significant downward trend since 2006. It might be caused by the global hotel group in China advocating green development. In 2013, the Chinese government issued regulations for practicing a strict economy and eliminating waste for the whole society, which caused a decrease in the supply scale of tourist hotels. To balance the demand and supply relationship and improve the revenue and profit, the hotel department further reduced the energy consumption in such links as power and heating system, catering and food, storage, and transportation, without affecting the customer experience [71].

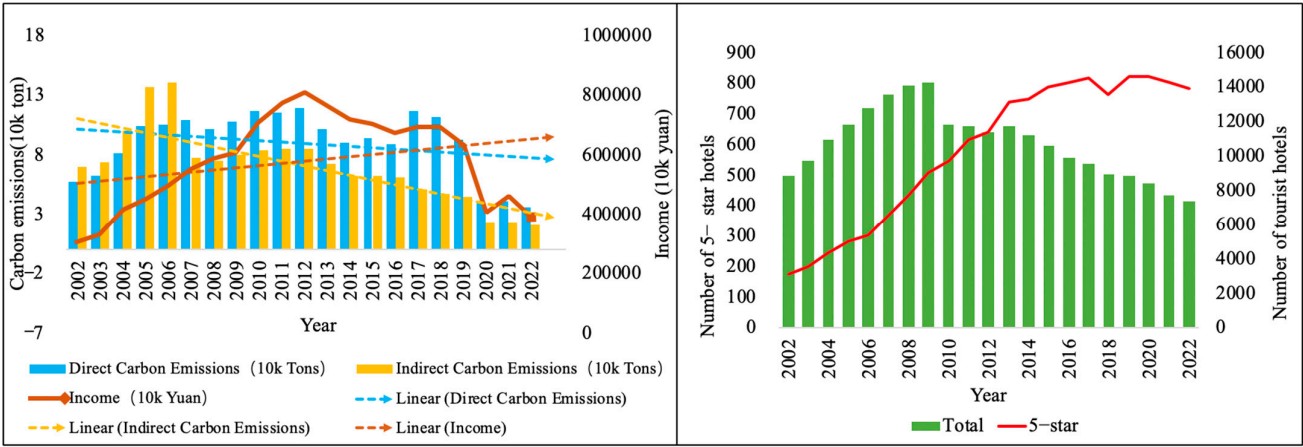

**Figure 1.** The average direct, indirect, and hotel revenue trends of Chinese tourist hotels from 2002 to 2022.

### 4.2. Characteristics of the Spatial Distribution of Direct Carbon Emissions and Indirect Carbon Emissions

The overall carbon emissions of tourist hotels in China show a spatial distribution characteristic with higher concentrations in the eastern and southern regions and correspondingly lower levels in the western and northern areas. The direct carbon emissions from tourist hotels along the southeast coastal region reached a peak over time before experiencing a decline, with this decreasing trend showing signs of extending into inland China (Figure 2). China's western region continues to witness an upward trend in direct carbon emissions. In contrast, the eastern region has experienced a more rapid decline in carbon emissions compared to the western region. Prior to 2014, the Yangtze River Delta and Beijing–Tianjin–Hebei regions stood out with notably higher indirect carbon emissions than other parts of the country (Figure 3). After that, the rapid decline of hotels' carbon emissions experienced an inverted U-shaped curve meant regional integration and a high degree of industrialization would help to reduce negative environmental externalities, and then promote the reduction of indirect carbon emissions in the hotel.

In 2002, there were a total of 8880 tourist hotels in China, with Guangdong Province having the largest number at 926 and Ningxia Autonomous Region having the smallest number at only 35. By 2022, there were a total of 7291 hotels, including 783 five-star hotels. From the average direct carbon emissions of each province based on the number of hotels, provinces such as Guizhou, Shanghai, Tianjin, and Xinjiang have the highest hotel carbon emissions, while hotels in Yunnan, Jiangxi, and Guangxi have an advantage in direct carbon emissions, and based on the number of hotels, there was a significant downward trend in direct carbon emissions from 2002 to 2022, which further confirms Section 4.1. From the perspective of the number of visitors to tourist hotels in various provinces, Shaanxi, Xinjiang, Shanghai, Hebei, and other provinces have the highest direct carbon emissions, and the emissions in 2022 were much higher than those in 2002. The average indirect and direct carbon emissions obtained from tourist hotels and hotel reception in each province are similar. It indicates that, especially in Xinjiang and Shanghai, the reception methods and business concepts of tourist hotels urgently need to be transformed.

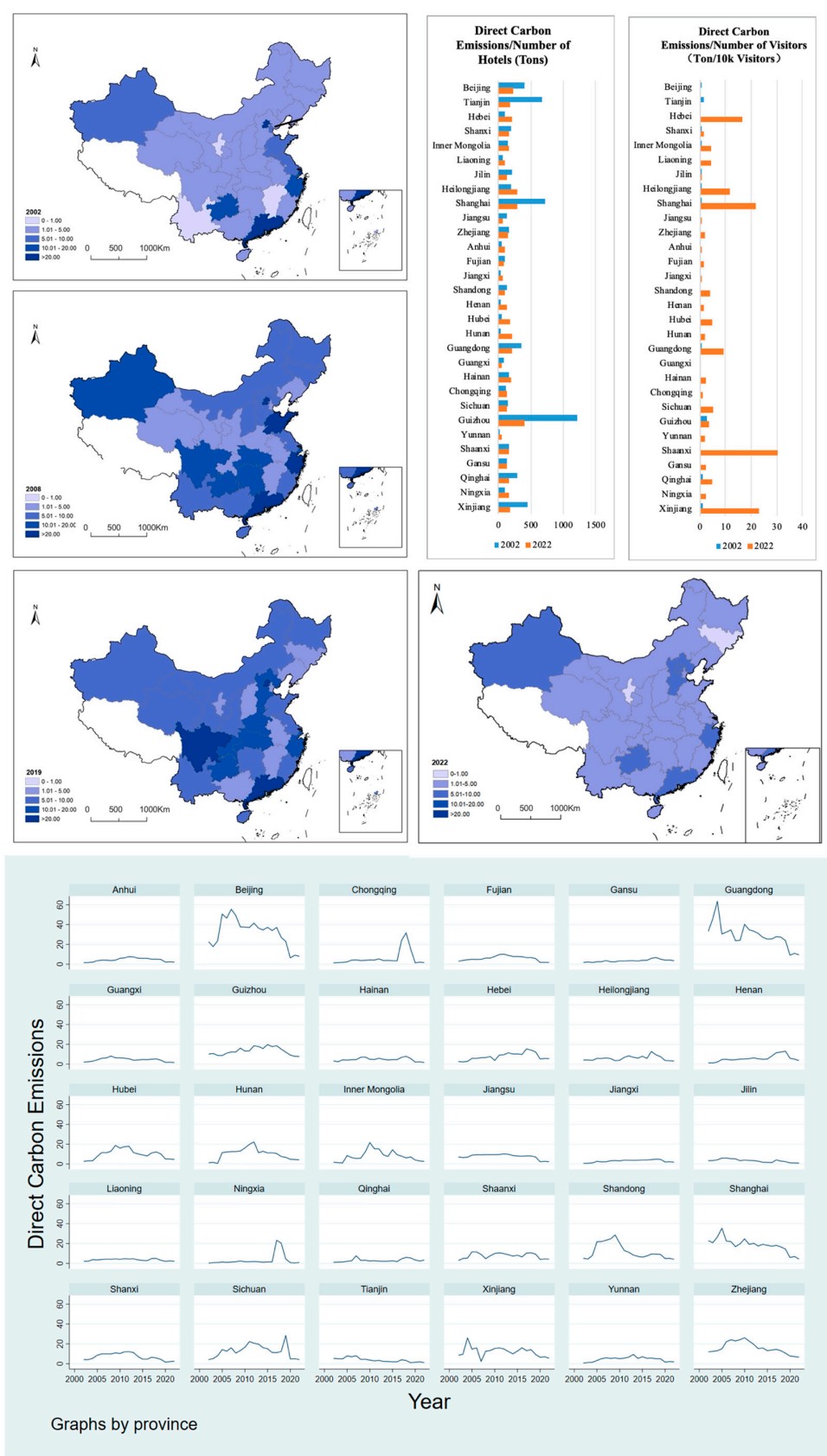

**Figure 2.** Spatial distribution and trend of direct carbon emissions of tourist hotels from 2002 to 2022.

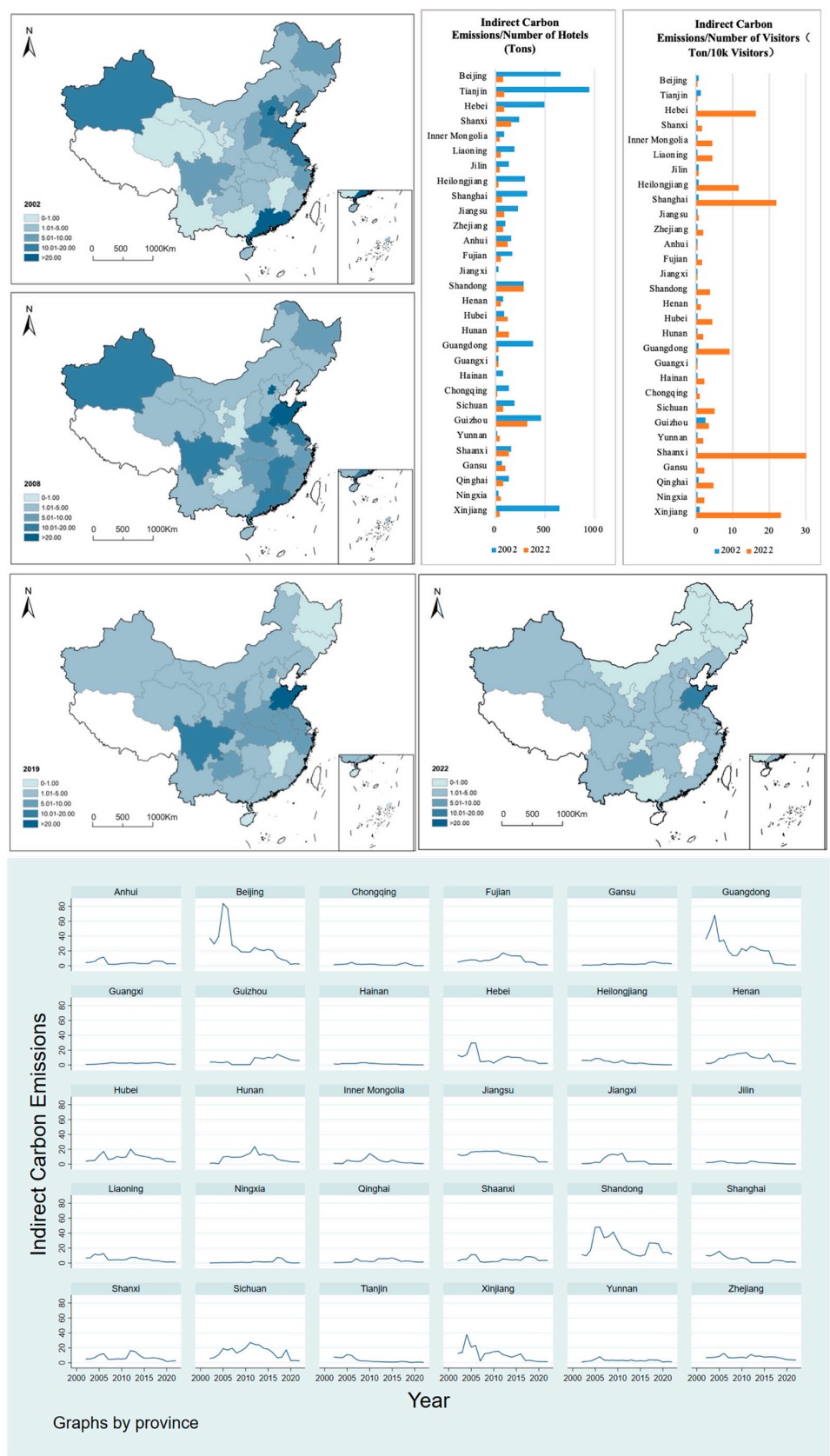

**Figure 3.** Spatial distribution and trend of indirect carbon emissions of tourist hotels from 2002 to 2022.

### 4.3. Indirect Carbon Emission Source Analysis

Tourist hotels of China had a kind of Engel's law effect, with the most indirect carbon emissions from the food and tobacco processing industry. Through Formula (7), the source sector of indirect carbon emissions can be measured and calculated, as shown in Figure 4. The food and tobacco processing industry was the largest contributor to the indirect carbon emissions of tourist hotels, accounting for 37.93% in 2002, rising to 47.58% in 2007, further increasing to 51.54% in 2015, and then slightly decreasing to 47.91% in 2017. It showed that the indirect carbon emissions of hotels had the 'Engel's Law' effect. Moreover, the carbon emission proportion within the food and tobacco processing industry initially rose and subsequently declined, displaying a trend akin to an inverted U-shaped curve. This reinforces the notion that the progression of high-standard tourist hotels is progressively steering the hotel industry towards a more efficient and environmentally friendly, low-carbon approach. Further, the industries with relatively high contributions to indirect carbon emissions are agriculture, forestry, animal husbandry and fishery, wholesale and retail trade, manufacturing, and rental services. In 2017, the carbon emissions contribution of the transportation sector increased by 53.18% compared to 2002. There are other service sectors with similar growth rates, mainly including waste management. The carbon emissions contribution in 2019 increased by 76.95% compared to 2012. These departments should attract timely attention and control from the government and relevant departments.

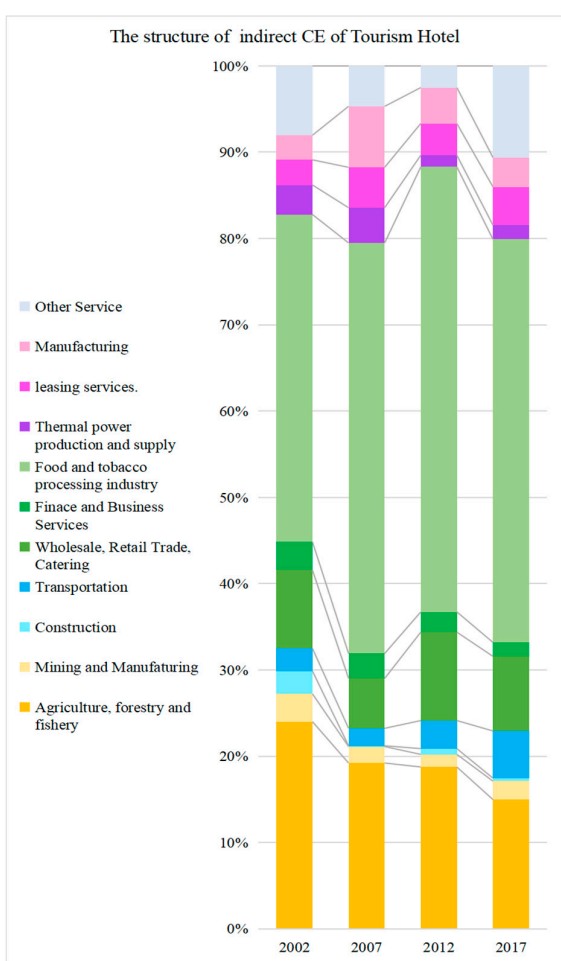

**Figure 4.** Composition of indirect carbon emission sources of tourist hotels.

### 4.4. Coupling Analysis of Carbon Emissions and Economic Value

China's 30 provinces were categorized into four distinct groups (Table 5), applying the average value of the income and carbon emissions, respectively (3.5 billion CNY of income and 60 k tons of carbon emission). The research coupled the income with direct and

indirect carbon emissions of tourist hotels, respectively, and conducted a scatter analysis of the variables (Figure 4).

**Table 5.** Characteristics and quantity of four types of Chinese tourist hotels.

| Coupling Relationship | Type | Characteristic | Quantity in 2002 | Quantity in 2022 |
|---|---|---|---|---|
| Income and Direct Carbon Emissions | Type I | Low-income, High-carbon emissions | 2 | 1 |
| | Type II | High-income, High-carbon emissions | 5 | 2 |
| | Type III | Low-income Low-carbon emission | 22 | 19 |
| | Type IV | High-income, Low-carbon emissions | 1 | 8 |
| Income and Indirect Carbon Emissions | Type I | Low-income, High-carbon emissions | 9 | 1 |
| | Type II | High-income, High-carbon emissions | 6 | 1 |
| | Type III | Low-income, Low-carbon emission | 15 | 19 |
| | Type IV | High-income, Low-carbon emissions | 0 | 9 |

The inclination of the linear regression for tourist hotels has diminished, indicating a gradual trend of decoupling (Figure 5). The slope of the regression line changed from 0.18 to 0.11 in the coupling analysis of direct carbon emissions and income. Most provinces steadily transitioned from type III to type II, and this transition may continue in the coming years. The number of first category provinces with low income but high emissions decreased from two in 2002 to one in 2022. Especially in Xinjiang, direct carbon emissions from tourist hotels should be promptly controlled.

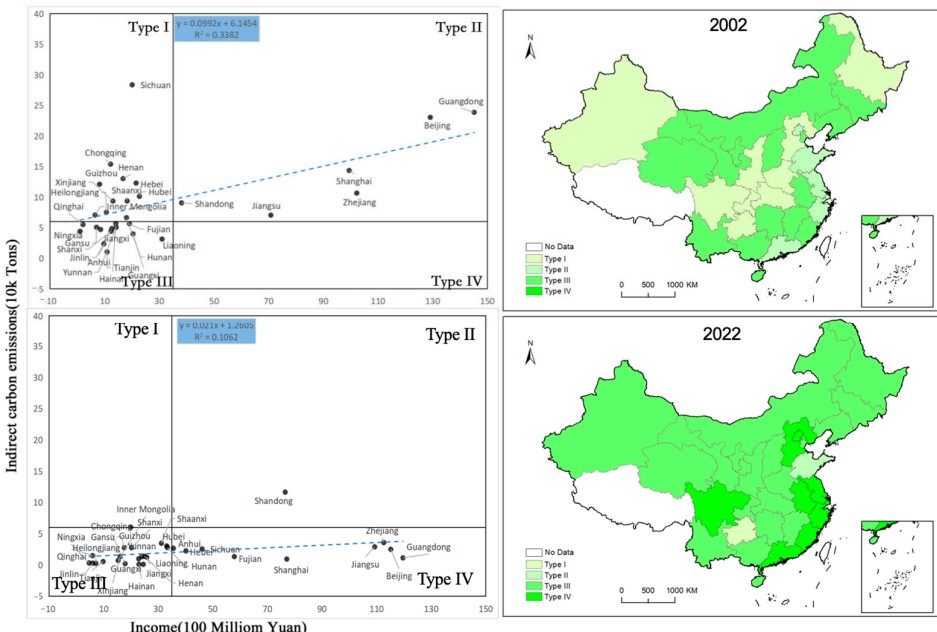

**Figure 5.** Coupling of income and carbon emissions.

China's tourist hotels exhibited a declining trend in indirect carbon emissions, with a projected decoupling from income growth, contributing to carbon peaking efforts (as illustrated in Figure 5). Between 2002 and 2022, the slope of the regression line for indirect

carbon emissions decreased from 0.10 to 0.02, demonstrating a more significant reduction in slope compared to that of direct carbon emissions. Type IV regions changed from zero to nine, such as Fujian, Zhejiang, Guangdong, and Shanghai in the coastal areas, and Henan and Hebei along the Yellow River. The carbon emissions of the above provinces began to decline after reaching a peak with the increase in income. It is worth mentioning that Beijing and Guangdong achieved a leap by transitioning from type II to type IV, considerably reducing indirect carbon emissions with increasing income. Beijing and Guangdong can be the model for the neighboring provinces.

### 4.5. Carbon Emission Efficiency of Tourist Hotels

The direct carbon emissions efficiency of tourist hotels is generally more than the total carbon emissions, showing a W-shaped curve. Figure 6 shows that the carbon emission efficiency fluctuated within ranges of 0.68–0.84 and 0.67–0.85 under different emission scenarios, respectively. Comparing this with Figure 1, it is evident that carbon emissions peaked in 2005 and 2006, a period characterized by relatively low levels of traffic and urbanization. Despite a high economic efficiency, these factors could not compensate for the high emissions, leading to a decrease in total carbon efficiency in 2006. This suggests that the adoption of low-carbon practices during the 2008 Beijing Green Olympics had a direct impact on reducing hotel carbon emissions. In 2010, the National Tourism Administration held the first large-scale national conference on improving the quality of tourist hotels, putting forward higher requirements for the operation scale, technology, energy consumption, and industrial structure of the hotel economic system, providing an important guarantee for the continuous rise of the hotel green development. Managers attached greater importance to the use of new technologies and constantly adjusted the management style, thus improving the overall eco-efficiency of carbon emissions in 2011 and 2012.

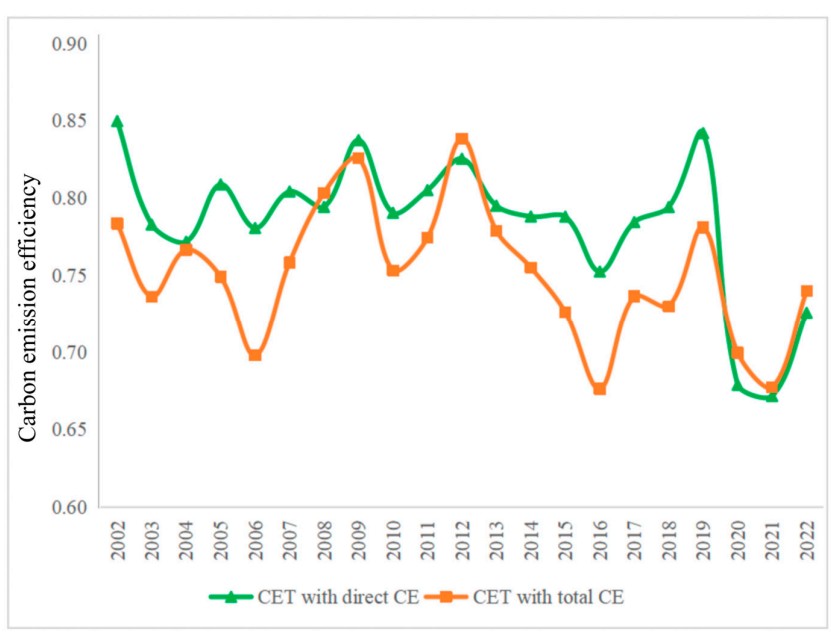

**Figure 6.** The efficiency of carbon emissions of tourist hotels in China.

### 4.6. Driving Factors Analysis

4.6.1. Analysis of the Internal Driving Factor Results

This paper uses the Tobit to show explore the driving factors that affect the carbon emissions and efficiency of Chinese tourist hotels under different conditions. The income index takes 2002 as the base period. Perform logarithmic was operated on variable data to avoid the regression result error caused by different dimensions of data [72]. The regression

results all passed the multicollinearity test and correlation test. The regression results of driving factors in different situations are explored in Table 6.

**Table 6.** Regression results of direct and indirect carbon emissions and its efficiency of China's tourist hotels.

| Type | Variable | DC | | IDC | | CET$_{direct}$ | | CET$_{total}$ | |
|---|---|---|---|---|---|---|---|---|---|
| | | Conf | z | Conf | z | Conf. | z | Conf. | z |
| Scale Effect | lnIncome | 9.09 | 35.76 *** | 7.30 | 13.80 *** | 0.04 | 2.38 ** | 0.05 | 2.95 *** |
| Technology Effect | lnEnergy | −8.03 | −28.93 *** | −5.15 | −10.45 *** | 0.17 | 9.38 *** | 0.13 | 6.41 *** |
| Structure Effect | TI | 0.17 | 6.87 *** | −0.11 | −2.32 ** | 0.00 | 2.09 ** | 0.00 | 2.48 ** |
| Regional Economic | lnGDP | −1.80 | −4.28 *** | −1.46 | −1.15 | 0.13 | 4.66 *** | 0.12 | 4.15 *** |
| Urbanization | UP | 0.03 | 1.35 | 0.01 | 0.21 | 0.00 | 1.17 | 0.00 | 0.90 |
| Water Resource | lnGPW | −0.19 | −1.19 | −1.16 | −3.98 *** | −0.03 | −2.59 *** | −0.03 | −2.67 *** |
| Energy Efficiency | GPE | −0.09 | −0.22 | −0.09 | −0.13 | 0.11 | 4.04 *** | 0.09 | 3.02 *** |
| Civilization | lnST | −0.77 | −2.45 ** | −0.44 | −0.74 | −0.16 | −7.54 *** | −0.20 | −8.87 *** |
| Traffic Conditions | lnHKM | −0.72 | −7.47 *** | 0.46 | 2.21 ** | 0.00 | 0.45 | −0.01 | −1.26 |

Note: **, and *** were shown to be significant at 0.05, and 0.01 levels. DC and IDC, respectively, represent the direct and indirect carbon emissions. CET$_{direct}$ and CET$_{total}$, respectively, represent the direct and total carbon emissions efficiency.

The scale effect represented by tourist hotel income has a significant positive correlation with carbon emissions and efficiency. Specifically, each 1% increase in income of the hotel will lead to an increase in direct and indirect carbon emissions of 9.09% and 7.30%, respectively, and the carbon emission efficiency under direct and total scenarios will increase by 0.04% and 0.05%, respectively, which showed that scale effect has a significant effect to improve carbon emission efficiency. However, the effect of increasing direct carbon emissions needs to be more attention. It is suggested that enterprises should adjust the scale appropriately. For instance, once the marginal returns reach their maximum point, adjustments can be made to the chain management practices to enhance the hotel's economic efficiency further, thereby compensating for any increase in carbon emissions.

An increase in hotel energy efficiency, measured by the unit energy income, results in a decrease of 8.03% in direct carbon emissions and a 0.17% boost in direct carbon efficiency for every 1% rise. This technological impact demonstrates its potential to mitigate carbon emissions and enhance efficiency across various scenarios. In the context of global environmental shifts, the tourism and hotel sector face an urgent need for technological advancements to pursue sustainable, low-carbon growth. To this end, hotels should incorporate cutting-edge technologies, such as smart air conditioning systems, energy-efficient room power systems, and intelligent management solutions, and allocate a greater portion of their initial investment towards technological enhancements [73]. This procedure will enhance energy efficiency and augment the additional advantages derived from investments in knowledge and technology, thereby fostering sustainable and efficient low-carbon development [74].

The structure effect represented by the proportion of 5-star hotels has a significant negative correlation with indirect carbon emissions, but positive correlation with the efficiencies—that was, high-star hotels—inhibit carbon emissions. Each 1% increase in the structural effect reduced the indirect carbon emissions of tourist hotels by 0.11%, which was consistent with the analysis in Figure 1 (With the increase in the number of 5-star hotels, there is a downward trend in indirect carbon emissions from hotels). It is because high-star hotels have formed mature low-carbon emission reduction ideas and introduced advanced emission reduction technologies, which have a suppressive effect on the direct carbon emissions in the hotel. The higher the hotel's star rating, the more attention is paid to emission reduction measures in connection with other departments, resulting in a reduction in indirect carbon emissions. However, due to the large scale of high-star hotels and the widespread phenomenon of high investment and high management costs, the quantity of high-star hotels showed a linear increase, while the income was not significant. It can

be seen that the demand scale of high-star hotels has not yet formed, and the operating efficiency of the economic system needs to be improved. It is suggested that high-star hotels can reduce labor or capital investment in the future. Moreover, it can expand the scale of demand, expand online social media marketing through internet new technology, and increase parent–child services, community activities, cultural experience, and other forms of business, to expand the market target population and improve comprehensive income capacity.

The regional economy, represented by the GDP, has a restraining effect on the hotels' carbon emissions, especially the direct carbon emissions; that was, every 1% increase in the economy reduced the direct carbon emissions by 1.80%, and it has a promoting effect on the its efficiency, indicating that the improvement of the overall economic can help to promote the realization of the "dual-carbon goal", which was consistent with the current status of China's hotel industry. The government departments should improve the regional economy, realize the reasonable concentration and optimal allocation of resource elements; promote the development of productivity and overall competitiveness; and levy environmental and resource taxes such as pollution charges and carbon taxes on enterprises with high energy consumption, emissions, and pollution.

The resources represented by the number of water resources per capita had a significant negative correlation with carbon emissions and its efficiency, indicating that the resource load in a region has a positive impact on tourist hotels but has not yet formed a good interaction mechanism for carbon efficiency. The reasonable use of tourist hotel resources can help reduce carbon emissions, but a good interaction mechanism has not yet been established for the improvement of economic efficiency. It is crucial to enhance the resource utilization efficiency of tourist hotels to improve carbon efficiency in the future.

Improving energy utilization has a mitigating influence on carbon emissions, but it is not significant in this regression. In previous studies, the sources of carbon emissions are mainly related to energy [75]. The use of new and clean energy can help reduce the carbon emissions of hotels. In the future, the hotel ought to facilitate the revision, transition, and enhancement of the energy mix via supply-side reforms, shifting from conventional energy sources like coal towards a diversified portfolio of clean energy, and foster sustainable and eco-friendly development practices.

The civilization, represented by the number of college students, has a negative correlation with the carbon emissions of tourist hotels. Every 1% increase in the degree of civilization has led to a reduction of 0.77% and 0.44% in carbon emissions of tourist hotels, respectively. Therefore, the improvement of the degree of civilization can help improve the overall quality of hotel occupants. In the future, the hotel should strengthen the national low-carbon theory education and promote the popularization of civilization education for the school-age population.

Each 1% increase in traffic represented by highway mileage can make decrease 0.72% direct carbon emissions and an increase of 0.46% in indirect carbon emissions. A large number of intermediate and close links with other industries in tourist hotels make the contribution the transportation to indirect carbon increase year by year, which is consistent with the analysis in Section 4.3. At present, China's tourist hotels are characterized by a significant spatial imbalance. High-star hotels in Beijing, Shanghai, Guangzhou, and other places account for one-third of the national hotels. In the future, the hotel should optimize the location of hotels in combination with traffic conditions and promote the coordinated development of regional tourism.

### 4.6.2. Analysis of External Driving Factor Results

This study selected five years (2002, 2005, 2010, 2015, 2019, and 2022) as the detection years. To explore the underlying factors influencing both direct and indirect carbon emissions from tourist hotels, we utilized the factor detection module and the interaction detection module of the geographical detector, respectively. The outcomes of this analysis are presented in Table 7.

**Table 7.** Results of the geographical detector factor detector model.

| | Year 2002 | | Year 2005 | | Year 2010 | | Year2015 | | Year2019 | | Year 2022 | |
|---|---|---|---|---|---|---|---|---|---|---|---|---|
| | Y1 | Y2 | Y1 | Y2 | Y1 | Y2 | Y1 | Y2 | Y1 | Y2 | Y1 | Y2 |
| X1 | 0.10 | 0.11 | 0.02 | 0.09 | 0.06 | 0.05 | 0.08 | 0.04 | 0.05 | 0.20 | 0.13 | 0.12 |
| X2 | 0.06 | 0.03 | 0.02 | 0.14 | 0.07 | 0.19 | 0.04 | 0.18 | 0.18 | 0.27 | 0.19 | 0.23 |
| X3 | 0.12 | 0.05 | 0.05 | 0.14 | 0.05 | 0.05 | 0.02 | 0.06 | 0.20 | 0.27 | 0.13 | 0.57 |
| X4 | 0.12 | 0.10 | 0.04 | 0.04 | 0.18 | 0.20 | 0.12 | 0.24 | 0.19 | 0.17 | 0.14 | 0.19 |
| X5 | 0.11 | 0.13 | 0.03 | 0.01 | 0.12 | 0.01 | 0.20 | 0.05 | 0.19 | 0.10 | 0.34 | 0.32 |
| X6 | 0.23 | 0.41 | 0.33 | 0.45 | 0.36 | 0.23 | 0.42 | 0.20 | 0.28 | 0.37 | 0.11 | 0.12 |
| X7 | 0.19 | 0.37 | 0.24 | 0.23 | 0.21 | 0.49 | 0.15 | 0.44 | 0.28 | 0.36 | 0.26 | 0.13 |
| X8 | 0.30 | 0.32 | 0.22 | 0.30 | 0.19 | 0.12 | 0.06 | 0.31 | 0.30 | 0.36 | 0.16 | 0.23 |
| X9 | 0.24 | 0.31 | 0.33 | 0.37 | 0.09 | 0.06 | 0.23 | 0.12 | 0.30 | 0.47 | 0.17 | 0.52 |
| X10 | 0.13 | 0.07 | 0.03 | 0.05 | 0.09 | 0.19 | 0.10 | 0.13 | 0.09 | 0.13 | 0.22 | 0.17 |
| X11 | 0.43 | 0.37 | 0.23 | 0.10 | 0.37 | 0.48 | 0.09 | 0.02 | 0.22 | 0.32 | 0.08 | 0.24 |
| X12 | 0.33 | 0.34 | 0.29 | 0.35 | 0.19 | 0.31 | 0.13 | 0.31 | 0.48 | 0.36 | 0.22 | 0.39 |

The results of single factor detector show that, in 2002, changes in consumer behavior represented by the number of domestic travel agencies were the main factors affecting the direct carbon emissions of tourist hotels, and the investment in environmental pollution control was the main factor affecting the indirect carbon emissions of tourist hotels. In 2022, area of protected zone and employed population were the main factors affecting the direct carbon emissions of tourist hotels, while average precipitation and the national technology market turnover was the main factor affecting the indirect carbon emissions of tourist hotels. Overall, the influence of natural factors on the direct and indirect carbon emissions of hotels is becoming more and more prominent, while that of human factors is diminishing. Therefore, when choosing the location of a hotel, natural factors should be taken into comprehensive consideration, and the interaction with human factors should also be considered.

In practice, the intricate supply chain of tourist hotels often means that the factors driving hotel carbon emissions are influenced by a multifaceted array of considerations. Therefore, based on the interaction detector results, it was found that the main factors are nonlinear enhancement and two factor enhancement. Among them, in 2002, the two interaction factors that played a major role in the direct carbon emissions of tourist hotels were the jurisdictional area and the total investment in environmental pollution control (Direct: q(X6∩X10) = 0.97; Indirect: q(X6∩X10) = 0.96), which represents nonlinear enhancement. In 2019, the two interaction factors that played a major role in the direct carbon emissions of tourist hotels were the forest coverage rate and overall daily average of energy-saving and emission reduction policy search index (q(X4∩X12) = 0.90), which showed non-linear enhancement. The forest coverage rate and national technology market transaction volume played a major role in the indirect carbon emissions (q(X4∩X9) = 0.92). Overall, the forest coverage area and human factors such as investment and policy constitute the primary elements influencing the carbon emissions of tourist hotels. Evidently, the augmentation of forest coverage exerts a positive inhibitory impact on carbon emissions. Nevertheless, if the combination of these factors with human-related investment and policies affects the carbon emissions of hotels, the process and mechanism of their influence represent significant directions for future studies on carbon reduction and carbon efficiency in hotels.

## 5. Discussion

While direct carbon emissions from tourist hotels demonstrate an upward trend, indirect carbon emissions exhibit a downward pattern. Prior research has indicated that China's tourism-related carbon emissions are increasing, with indirect emissions significantly outpacing direct emissions [76]. However, our study reveals a rapid decline in indirect carbon emissions from tourist hotels, to the point where they are now lower than direct emissions. Our findings indicate that China's tourist hotels hold the promise of leading the way in

decoupling efforts through ongoing improvements in their supply chain management, ultimately aiming to refine and regulate tourism-related carbon emissions. Various initiatives and strategies, exemplified by guidelines promoting sustainable and robust growth in the tourist hotel industry, exert a favorable influence on these establishments. This aligns with the research conducted by Mohammed Benlemlih, which underscores the efficacy of governmental policy support in mitigating carbon emissions [77].

When considering indirect carbon emission, high-star hotels are not the main carbon emission department. Zi Tang also proved that low-star hotels are the largest carbon emission departments rather than high-star hotels [27], which is consistent with the research in this study. The key contributor to indirect carbon emissions from hotels originates from the sectors of food production and tobacco manufacturing. The contribution rate of these industries to indirect carbon emissions follows an inverted U-shaped curve, aligning with the findings of Viagaslau Filimonau's study on greenhouse gas sources in British hotels [20]. This also corroborates Boyka Bratanova's research on the rise in customers' food consumption [78]. It shows that there is also the 'Hotel Engel's Law' in the tourism and hotel industry; that is, with the increase of the proportion of leisure and holiday tourism, the carbon emissions of food consumption appear in an upended U-shaped curve that first increases and then decreases. At the same time, it also coincides with Milindi's research that the disposal of leftover food in restaurants can lead to significant amounts of indirect greenhouse gas emissions [79].

## 6. Implications

The tourist hotel should focus on improving its technical effect, while it also should focus on improving the regional economy. For the interior of the tourist hotel system, new technologies should be introduced and carry out a thorough technological revolution without affecting the experience of the guests. Ahmed and others researched five South Asian countries, and the findings showed that a causal link existed between technological economic factors and the level of carbon emissions [80]. Outside of the tourist hotel system, the high economy can indirectly control the carbon emissions. Dong et al. also found that transformations in the economy and advancements in regional development can serve as key strategies for a particular industry to cut down on carbon emissions [81], which confirmed the research in this paper. Through the joint efforts of the hotel system both internally and externally, Chinese tourist hotels possess the potential to spearhead the tourism sector in meeting the dual carbon targets, holding substantial importance for China's efforts to advance the global vision of a shared future for mankind and the achievement of sustainable development objectives.

It is suggested to promote regional coordinated development while formulating different carbon reduction measures according to local conditions. The linkage effect of the southeast coastal zone on other regions should continue to give play to the spillover effect and establish a coordinated development mechanism. For example, on the premise of protecting customer privacy, the hotels can establish a customer-sharing system, launch preferential joint tickets, and establish regular employee joint training and exchange activities to enhance the professionalism and enthusiasm of hotel staff and adopt different energy measures for provinces with significant geographical differences. There are significant differences in altitude, temperature, precipitation, and other indicators between Xinjiang and Hainan. Hainan has a tropical monsoon climate and abundant resources. The annual average temperature is 22–26 °C, and the annual average precipitation is 1639 mm. These advantages should be utilized to launch eco-friendly hotels, generate electricity using solar and tidal energy, and use forest and water resources as important carbon sink tools. Xinjiang boasts a temperate continental climate characterized by substantial temperature variations, abundant sunshine, minimal precipitation, and arid conditions. It is imperative to expedite the advancement of wind and solar power generation while actively exploring the potential of biomass energy resources and achieve balanced development of energy structure.

It is necessary to strengthen low-carbon awareness and guide tourist hotels to achieve decoupling goals. Enterprises should rectify and transform the internal high-energy consumption part, improve their independent innovation ability, and regularly measure carbon emissions and publish them to the public; encourage local communities to participate; and thus enhance the supervision function of the whole society. Additionally, hotels can also guide the concept of low-carbon consumption of the whole people through media, forums, and short videos to encourage customers to improve their awareness of energy conservation, such as advocating low-carbon catering, reducing the use of disposable tableware, scientific use of electricity and water, and checking whether the air conditioning is turned off before leaving the hotel. In addition, it is crucial for Chinese tourist hotels to achieve green transformation in the future through carbon reduction policies that reduce energy consumption and harmful emissions. The gap between customers' attitudes and behaviors towards the environment when using landscape plants in hotels can affect the hotel's emissions and profits, especially during the off-season [82], Plant materials can help hotels fix carbon emissions and improve environmental quality [83], Avoid proactively providing disposable household items while ensuring guest satisfaction. Especially low-income and high carbon emitting hotels should actively use the above policies to achieve green transformation of Chinese tourism hotels this morning.

Regional governments should support the construction of low-carbon hotels and build an ecosystem of low-carbon hotel chains. Through establishing a strategic partnership with enterprises in all links upstream and downstream, enterprises should make full use of various resource advantages to drive the supply chain industry to jointly realize the carbon emission decoupling strategy. The government should precisely regulate the food and tobacco industry, transportation industry, and postal storage industry in the supply chain of tourist hotels and strive to take the lead in achieving the goal of carbon emissions decoupling. For example, the government and enterprises can encourage customers to reduce the purchase of over-processed food, implement the food carbon labeling policy, reduce storage energy consumption, encourage the hotel to use local materials, and avoid long-distance transportation.

It is worth noting that, at the provincial level, tourist hotels in Zhejiang Province are a national example in low-carbon and energy-saving. This is consistent with previous research [84]. It is because Zhejiang launched the "Creating Green Hotels and Advocating Green Consumption" campaign in the entire industry as early as 1999. By the end of 2023, Zhejiang Province has created a total of 396 green tourist hotels, ranking first in the country in terms of total number. Specifically, energy efficiency has decreased from 36.6 kg of standard coal in 2009 to 18.2 kg of standard coal in 2022. The use of energy sources such as electricity and natural gas has become widespread in tourist hotels, while diesel and coal have been largely phased out. In terms of water resource utilization, the average total water consumption of tourist hotels has continued to decline, with a decrease of 33% from 2016 to 2020. Due to the one-time consumption cost of tourist hotel management and service by the Ministry of Culture and Tourism in 2021, Wenling International Hotel was selected as a typical case of star-rated hotel management and service by the Ministry of Culture and Tourism in 2021, setting a learning example and demonstration for the high-quality development of the national hotel industry. Wenling International Hotel is based on the entire industry chain and implements energy-saving and emission reduction measures throughout the entire life cycle, involving various fields such as energy-saving buildings, energy-saving renovations, energy-saving equipment procurement, low-carbon marketing, and energy-saving management. In terms of promotion targeting the consumer end, Wenling International Hotel in Zhejiang Province has deeply implemented the action of using public chopsticks and spoons and stopping food waste, creating a strong atmosphere of promoting thrift and shame in waste. Deepen the promotion of plastic pollution control actions and guide guests to bring their own toiletries.

## 7. Determinants and Future Studies

Direct carbon emissions exhibited an upward trend, whereas indirect carbon emissions demonstrated a downward trend. The spatiotemporal distribution of carbon emissions among tourist hotels was highly uneven, attributed to the combined influence of socioeconomic and natural factors. Notably, hotels in most western provinces experienced rising carbon emissions, whereas those in eastern provinces displayed a U-shaped pattern, mirroring the environmental Kuznets curve. The main contributors to indirect carbon emissions from China's tourist hotels were the food processing and tobacco manufacturing sectors, hinting at a possible adherence to Engel's law within the hotel industry. Considering the number of tourist hotels and hotel receptions per province, Xinjiang and Shanghai emerged with the highest average carbon emissions, highlighting the urgency for these regions to transform their business models and concepts.

The annual decline in the slope of the regression line for tourist hotels in China indicates a gradual decoupling trend. The direct carbon emissions in most provinces increased with the increase in income, indicating that tourist hotels have not yet achieved high-quality development, while the indirect carbon emissions with the supply chain continue to decrease. Moreover, the slope of the regression line under indirect carbon emission scenarios changes greatly, indicating that China's tourist hotels were anticipated to undergo a significant shift, transitioning from low-income levels with indirect carbon emissions to higher-income statuses, as well as may lead to achieving the carbon peak through reducing indirect carbon emissions.

The direct carbon emission efficiency of tourist hotels in China is generally better than total carbon emission, showing a W-shaped curve. The technological advancements and regional configurations exert a favorable influence on mitigating carbon emissions in the hotel industry. Therefore, the internal of the tourist hotel should focus on improving the technology effect, and the external should focus on improving the regional economy specifically.

Adopting energy-saving technology can enhance the carbon performance of hotels, while regional economic growth can benefit the carbon emissions efficiency indirectly. The scale effect represented by tourist hotel income has a significant positive correlation with carbon emissions and efficiency. The proportion of 5-star hotels, serving as an indicator of structural impact, exhibits a notable inverse relationship with both carbon emissions and their efficiency, particularly in terms of indirect carbon emissions. Furthermore, energy efficiency plays a crucial role in mitigating carbon emissions.

The innovation of this study lies in exploring the driving factors that affect carbon emissions and efficiency and proposing a more targeted green decoupling development strategy for tourist hotels. A research framework for carbon emissions has been constructed, providing a new method for measuring the coordination between the economy and ecology of tourist hotels. Long-term carbon emissions have also been studied, providing comparability for research on short time series and small spatial scales. Environmental pollution control investment serves as an external stimulus influencing direct carbon emissions from tourist hotels, whereas the workforce constitutes an external factor impacting indirect carbon emissions in this sector.

The limitations of this research are the constraint posed by the accessibility of data when assessing the carbon emissions of tourist hotels in China, the research period of this study is limited to 2002–2022, and there is a lack of sufficient long-term research exploring the relationship between carbon emissions and the economy of tourist hotels in the post-pandemic era. This is also a key focus that needs to be supplemented and paid attention to in the future. In the future, this study will update the latest data, track the latest method models, and conduct more in-depth research.

**Author Contributions:** Conceptualization, B.X.; software, J.Z.; data curation, J.Z.; writing—original draft, J.Z.; writing—review and editing, B.X.; funding acquisition, B.X. All authors have read and agreed to the published version of the manuscript.

**Funding:** This work was funded by the National Natural Science Foundation of China (42201321).

**Institutional Review Board Statement:** Not applicable.

**Informed Consent Statement:** Not applicable.

**Data Availability Statement:** The data will be provided on request.

**Acknowledgments:** The authors would like to thank the editor and reviewers for their insightful comments and suggestions.

**Conflicts of Interest:** The authors declare no conflicts of interest. The authors affirm that they have no financial conflicts of interest or personal relationships known to them that might be perceived as having influenced the research presented in this paper.

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
