# Peer review of "Carbon Emissions and Its Efficiency of Tourist Hotels in China from the Supply Chain Based on the Input–Output Method and Super-SBM Model"

_sustainability, doi:10.3390/su16219489_

Round 1
Reviewer 1 Report
Comments and Suggestions for Authors
Studies had shown that tourism's carbon emissions are for up to 8% of the global total carbon emissions and grow at a rate of 3%.These data are worrying if you consider that hotels and resorts exceeded $ 1 trillion annually from 2016 to 2019, and they have rising trend. 1% of global total carbon emissions made by hotels and that percentage has also a rising trend. China contributed 28.8% of the world's carbon emissions in 2019, which justifies the research topic of this paper. The carbon emissions of the accommodation sector is a question that troubles hoteliers in other parts of the world as well. The Literature Review as well as References cited in the manuscript are appropriate and relevant to this topic as well as the Super-SBM model which is used to evaluate the eco-efficiency of tourist hotels. Calculation of Carbon Emissions with explanations of calculation on Direct Carbon Emissions are clearly explained.
I believe that the paper would be good to publish if there was not a large time gap from 2019 to 2024. The conclusions drawn on the basis of statistical-mathematical analysis, accompanying diagrams and pictures are good, but refer to the period 2002-2019.
The authors emphasized in the last sentence of the 6th part: “The comparative analysis of carbon emissions and efficiency of tourist hotels before and after the outbreak of the epidemic will also be the focus of future research.”, but I think that this is a big shortcoming of this manuscript and that the research should have covered the period at least until 2022.
Everything written is fine, it just needs to be extended in time.
Reviewer 2 Report
Comments and Suggestions for Authors
Even after the introductory section, it is clear that this work is an important study of the environment of one of the largest countries in the world. Tourism and the hotel industry, as a growing part of the Chinese economy, attracts a large number of foreign and domestic tourists, causing significant carbon emissions. The entire work relies on a high-quality database and appropriate methodology, which has produced verified data that provides insight into the distribution of carbon emissions from hotels in China based on multiple indicators, which is certainly good.
However, I must note that nowhere in the paper is the number of hotels per province analysed as the basic unit of carbon emissions.
Nor were the statistics on tourist arrivals and hotel capacity utilisation during the period under study analysed anywhere. After all, it is the tourists who ensure the existence of the hotels and cause carbon emissions to rise or fall.
In this context, I have recommended that the authors supplement their work with data, analyses and possible results at certain points in the revised version.

Reviewer 3 Report
Comments and Suggestions for Authors§ Thank you for the opportunity to review the paper. The paper aims to provide a scientific framework for the low-carbon development of the accommodation industry in China and similar regions in the world. This paper also reviewed the EEIO (Environmentally Extended Input-Output) model and the Super-SBM (Slack-Based Measurement) model to measure carbon emissions and their efficiency including indirect carbon emissions from the supply chain in China from 2002 to 2019. This paper has originality because it fills the gap in existing studies that have largely focused on analyzing the direct carbon emissions65 or their drivers in tourist hotels without noticing the carbon emissions from the supply chain67 and the economic value part of tourist hotels. The paper relied on bridging the gap by achieving six main objectives, namely, calculating the efficiency and trends of direct and indirect carbon emissions of the tourism hotel supply chain in China from 2002 to 2019, visualizing direct and indirect carbon emissions to explain their spatial distribution, identifying the main sources of indirect carbon emissions from tourism hotels, exploring the coupling relationship between carbon emissions and the economic value of Chinese tourism hotels, analyzing the carbon efficiency of hotels under direct and indirect carbon emissions scenarios, and exploring the driving factors affecting carbon emissions and efficiency of Chinese tourism hotels.
§ Introduction: It is suggested to update the ratios mentioned in the first paragraph because it was based on a reference from 2018. The introduction also needs to add more details about the role of tourism hotels in carbon emissions through tourism activities. The introduction also needs to clarify the gap more deeply, and highlight the theoretical and scientific contributions of the paper.
§ Literature Review: It needs to expand on the review of previous recent studies that have addressed the relationship between carbon emissions and their impact on the efficiency and effectiveness of hotel performance. For the model; It needs to add some details about how the model works, what its outputs are, and how it works in hotels.
§ Methodology: I suggest adding a separate section at the beginning that explains the methodology on which the paper relied, its justifications, and how it is suitable for application to hotels. I also suggest clarifying the reasons for choosing the mentioned calculation methods, and why they were preferred over other methods. The tables also need more explanation and clarification to be easier to read.
§ Results and Analysis: I suggest clarifying some results related to the efficiency of measurement reasons in reaching accurate results about calculating carbon emissions, and the extent of their agreement or difference from carbon emissions from other sectors or activities or other countries.
§ Discussion and Implications: I suggest separating the implications from the discussion, and focusing on how to reduce carbon emissions, and how to encourage hotels to shift towards green hotels and sustain all their activities.
§ Conclusion: I suggest changing it to determinants and future studies, and stating if there are updates related to the methodology, analysis, or data availability.
Comments on the Quality of English LanguageMinor editing of English language required
Round 2
Reviewer 1 Report
Comments and Suggestions for Authors
I thank the authors who appreciated recommendations and made the first version of the paper even better. New study applied the regional input-output table in a mathematical and statistical research to calculates the direct and indirect carbon emissions including the supply chain of Chinese tourist hotels from 2002 to 2022. The research period was from 2002 to 2019 in previous version of the proposal paper. As a result the revision includes updating the data of tourism hotel related indicators, including direct, indirect and complete carbon emissions, carbon emission efficiency under multiple scenarios, driving factors in Tobit regression, and geographic detector processing, which have been updated to 2022. As result I think that the content succinctly described and contextualized previous and present theoretical background and empirical research on the topic; the research design, questions, hypotheses and methods are clearly shown; the arguments and discussion of findings are balanced and compelling; empirical research are clearly presented. As previous paper, this one also uses references.
In general, the revised paper satisfies all the previous requirements, especially the extension of the time series to 2002-2022, so I suggest that the paper to be accepted in a present form. I congratulate the authors on an excellent manuscript, by my opinion.
Reviewer 2 Report
Comments and Suggestions for Authors
The authors made significant changes, and responded to the suggestions I made. The work is now enriched with data that indicate carbon emissions in China in more detail at the regional level.
Reviewer 3 Report
Comments and Suggestions for Authors
All suggestions are reflected in the paper with high accuracy. The paper is clearer to the reader after adding more details. I have no further questions or suggestions.